# Developing a sustainability strategy for Taiwan's tourism industry after the COVID-19 pandemic

**Ming-Chi Tsai**  *

Department of Business Administration, I-Shou University, Kaohsiung City, Dashu District, Taiwan

* mct@isu.edu.tw

## Abstract

The outbreak of COVID-19 around the world has caused great damage to the global economy. The tourism industry is among the worst-hit industries. How to focus on visitors who are most helpful to the tourism industry and develop sustainable strategy of operation is a very important question for after the epidemic is over. This study applied two-stage data envelopment analysis (DEA) and principal component analysis (PCA) to investigate past statistics from the Tourism Bureau and explore the shopping patterns of tourists who travel to Taiwan. The focus will be on tourists from major countries such as China, Japan, and Southeast Asian countries. According to the analysis of tourists from different countries, the money spent by tourists from different countries is concentrated on different items, and there are subitems that they particularly like to purchase. For the analysis of the purpose of coming to Taiwan, some tourism areas worth developing (such as medical treatment and leisure) are also presented in the research results. Based on these results, and according to the sustainable development goals, specific recommendations for the sustainability strategy of operation are made as a reference for the government and relevant industries. This research also broadens the scope of application of DEA and points out a different direction for future research.

## Introduction

The United Nations World Tourism Organization (UNWTO) estimates that there were just 25 million international tourist arrivals in 1950. This number increased to 1.4 billion international arrivals in 2018, an increase of 56-fold in 68 years. The number of international tourists in the Asia and Pacific region has increased from 200,000 in 1950 to 343,000,000 in 2018 [1]. The global tourism market shows a long-term stable growth trend, with an average annual growth rate of about 3.3%, which is slightly slower than the 3.9% from 1995 to 2010 [2]. In addition, according to the statistics of the World Travel & Tourism Council (WTTC), 1 out of 11 people was employed in the tourism industry in 2015 and the total number of employees in the tourism industry exceeded 284 million people, contributing more than 7.2 trillion USD to the global GDP. The total number of employees has exceeded the population of Brazil.

**Data Availability Statement:** Data are available from the Tourism Bureau (https://admin.taiwan.net.tw/BusinessInfo/TouristStatistics).

**Funding:** The author received no specific funding for this work.

**Competing interests:** The author have declared that no competing interests exist.

Tourism accounts for about 10% of the global GDP [3]. The WTTC mentioned that, in 2018, tourism contributed US$8.8 trillion to the global economy, accounting for 10.4% of all economic activities. The WTTC estimates that tourism accounts for 319 million jobs worldwide [4]. International tourism is undergoing rapid growth, as all countries are making every effort to strengthen tourism resource development and marketing to attract more international visitors and increase tourism income [5].

Since the outbreak of a new coronavirus (COVID-19) around the world, many countries have a large number of people infected by this new virus. By January 3, 2021, the number of confirmed cases reached more than 84 million globally and deaths numbered more than 1,834,000 [6]. Many governments have had to lock down cities so as to prevent the virus spreading to more people. Most flights between countries have been canceled, too. The global economy has suffered hugely, and the unemployment rate is increasing quickly. In the United States, the unemployment rate reached 14.7% in April 2020, compared to 3.5% at the end of 2019 [7]. The WTTC warned that 100 million people in the global tourism industry might be unemployed this year. The UNWTO survey indicates that there are now restrictions on nearly 100% of destinations, of which 83% of regions have implemented limits on tourism for more than four weeks. The WTTC also warned that the epidemic has reduced employment in the tourism industry by about 100 million, of which nearly 75 million jobs are located in G20 countries [8].

Many industries are unable to continue production due to work stoppages, resulting in worker unemployment and even bankruptcy. Tourism is among the worst-hit industries. Most people around the world have stopped traveling or dining out. Taiwan has done a good job of epidemic prevention, with only a small number of people infected. However, because most of its income depends on international tourists, Taiwan's tourism industry has still been greatly affected. We do not yet know when this pandemic will end. After the epidemic is over, the tourism industry will want to attract international tourists to visit Taiwan as soon as possible. At that point, how to focus on the tourists who will be most helpful to economic recovery, so that the industry can recover in the shortest possible time, is a very important question. Currently, many studies are focusing on COVID-19, including medical aspects [9–16] (transmission, symptoms, treatment, etc.) and political and economic influences [17–22]. Only few researches are focus on the impact of tourism industry [23–25].

The United Nations Member States adopted "The 2030 Agenda for Sustainable Development" in 2015 to provide a shared blueprint for peace and prosperity for the world. At its heart are the 17 Sustainable Development Goals (SDGs), which are an urgent call for action by all countries in a global partnership [26]. Among the 17 Sustainable Development Goals, the 8th, 9th, 12th, and 17th are "decent work and economic growth," "industry, innovation, and infrastructure," "responsible consumption and production," and "partnerships for the goals," respectively. According to these goals, the main objective of this study is to find a sustainability strategy of operation for Taiwan's tourism industry.

Current studies about the impact of COVID-19 to tourism industry still lack of quantitative study [23–25]. To reach above objective, the research question of this study is how to combine the concept of efficiency evaluation and the contribution of tourists to tourism industry and hence develop sustainable operation strategy for the tourism industry in Taiwan. Theoretically, this study can fulfill the quantitative research part of tourism industry and extend the application of efficiency evaluation techniques.

Using two-stage data envelopment analysis (DEA) and principal component analysis (PCA), this study investigates past statistics and explores following results:

1. Major international tourists are from China, Japan, Korea, and several Southeast Asian countries.

2. In the past five years, travelers from Japan spent most money on hotel, dining, and transportation expenses. In terms of shopping, they prefer "Famous product or specialty" and "tea".

3. Travelers from China spent more money on shopping than other countries. They spent the most money on "Jewelry or jade" followed by "Famous product or specialty".

4. Tourists from countries that are willing to spend more on accommodation are also willing to spend more on catering and entertainment. On the other hand, travelers who are willing to spend more on shopping generally also spend more on transportation.

5. For tourists who visit Taiwan for different purposes, travelers who spent more on hotel expenses also spent more on dining and transportation. On the other hand, travelers who spent more on shopping also spent more on miscellaneous expenses.

6. Travelers who spend more on any one subitem in "Jewelry or jade", "Tobacco or alcohol", "Electronic or electrical appliances", "Cosmetics or perfume", and "Clothing or related accessories" also spend more on the other four subitems.

Based on above results, specific recommendations for a strategy of sustainable operation are made with reference to the relevant industries.

## Related works

### The overall trend of Taiwan's tourism market

Taiwan's tourism development is based on the establishment of the "Taiwan Provincial Tourism Commission" in 1956 [27]. The most rapid development started in the 1970s. According to the Tourism Bureau, there were 512,776 tourists visiting Taiwan in 1972; 1980 was the year with the largest number of tourists visiting Taiwan in the past, at 1.3 million people. Despite the impact of the Asian financial crisis and the earthquake in 1999, the number of tourists who visited Taiwan in 2001 still reached 1,021,572. The Executive Yuan proposed the "Taiwan Double" in 2002, and the number of visitors to Taiwan for the purpose of tourism in the plan's goal was increased from 1 million in 2001 to more than 2 million. The focus of the tourism doubling plan was to increase the demand for the domestic tourism market and activate the job market to slow down the unemployment rate [2].

In general, the total number of domestic tourist trips in the country has been increasing in the past decade, from 97.99 million in 2009 to 171.09 million in 2018. Although there have been several drops in the process, this change is relatively small. This also shows that the government has actively promoted tourism in recent years, including domestic tourism, which should have a positive effect in terms of enhancing Taiwan's tourism output value and economic benefits [2].

In the past 10 years, the number of people visiting Taiwan has increased from 5,567,277 in 2010 to 11,864,105 in 2019. The number of trips abroad also increased, from 9,415,074 in 2010 to 17,101,335 in 2019 [2]. The number of people going abroad is even higher than the number of tourists coming to Taiwan. To sum up, Taiwan's government strategy not only aims at developing domestic tourism but also actively promotes international tourism, with the aim of building Taiwan into an island of tourism. If this momentum can be maintained, it will contribute to the future development of the tourism industry and economic growth.

Taiwan is strategically located in the center of East Asia. In addition, it has special natural landscape, food, and cultural characteristics, and has international-level tourism environment conditions. How to attract international tourists to visit Taiwan to increase domestic tourism employment opportunities and foreign exchange earnings from tourism, as well as enhance

**Table 1. The number of tourist visits to Taiwan in 2019.**

| Residence | Total | Business | Leisure | Others |
|---|---|---|---|---|
| Hong Kong, Macao | **1,758,006** | 84,243 | 1,527,072 | 146,691 |
| China | **2,714,065** | 15,935 | 2,052,401 | 645,729 |
| Japan | **2,167,952** | 250,285 | 1,680,682 | 236,985 |
| Korea | **1,242,598** | 54,970 | 1,040,352 | 147,276 |
| India | 40,353 | 11,005 | 5629 | 23,719 |
| Middle East | 24,030 | 7577 | 7994 | 8459 |
| Malaysia | **537,692** | 21,885 | 402,392 | 113,415 |
| Singapore | **460,635** | 48,451 | 352,510 | 59,674 |
| Indonesia | 229,960 | 5231 | 59,428 | 165,301 |
| Philippines | **509,519** | 9239 | 306,660 | 193,620 |
| Thailand | **413,926** | 11,784 | 300,352 | 101,790 |
| Vietnam | **405,396** | 7515 | 144,589 | 253,292 |
| Others | 57,567 | 3178 | 7592 | 10,533 |
| Asia Total | 10,561,699 | 532,315 | 7,907,366 | 2,122,018 |
| United States | **605,054** | 101,361 | 231,156 | 272,537 |
| Others | 161,200 | 12,140 | 82,477 | 66,583 |
| Americas Total | 766,254 | 113,501 | 313,633 | 339,120 |
| Europe Total | 386,752 | 86,240 | 151,949 | 148,563 |
| Oceania Total | 134,860 | 11,156 | 68,720 | 54,984 |
| Africa Total | 12,537 | 2799 | 1829 | 7909 |
| Unstated | 2003 | 104 | 527 | 1372 |
| Grand Total | 11,864,105 | 746,115 | 8,444,024 | 2,673,966 |

Source: [2].

Taiwan's overall economic growth and expand its international reputation, has been a key goal of the government's long-term efforts.

The total revenue of international tourism had never exceeded three billion USD before 2000 in Taiwan. However, the total revenue of international tourism increased from US$5.936 billion in 2008 to US$14.411 billion in 2019 [28]. It has become a key service industry in Taiwan in recent years. Since 2008, Taiwan's international tourism income has surpassed its domestic tourism income. The target market representing Taiwan's tourism industry has moved from the domestic market to the international market. This also proves that the tourism industry is a smoke-free environmental protection industry that can generate foreign exchange income.

Table 1 shows the number of tourists to Taiwan in 2019. It can be seen from Table 1 that China, Japan, Hong Kong, Macao, and Korea are the main countries or regions with more than one million visitors per year. This is followed by the United States, Malaysia, the Philippines, Singapore, Thailand, and Vietnam, etc., whose number of tourists ranged from 400,000 to 600,000 per year.

Since the outbreak of COVID-19 in early 2020, Taiwan has closed routes for most Chinese cities and implemented border controls, which has caused a significant drop in the number of Chinese tourists. At the same time, various countries have also implemented border controls one after another, causing the number of international visitors to Taiwan to drop significantly from January to May 2020. The total number of tourists in the first five months of 2020 was 1,254,395, only about one-tenth of the number from the previous year [2].

Even though Taiwan has done a good job of epidemic containment, with all economic activities maintained normally, schools starting classes normally, and no companies or factories shutting down due to the pandemic, Taiwan's tourism industry has still been affected hugely because most of its income depends on international tourists.

New vaccines are gradually developed, so hopefully the pandemic will end within months. However, are international travelers coming back so quickly as expected and maintaining the same consumption behavior? Therefore, how to attract international tourists to visit Taiwan again as soon as possible is a very important question, and the country must focus on those tourists who are most helpful to their operational performance so that the industry can recover in the shortest time possible.

## The 2030 agenda for sustainable development

The United Nations Member States adopted "The 2030 Agenda for Sustainable Development" in 2015 to provide a shared blueprint for peace and prosperity for the world. At its heart are the 17 Sustainable Development Goals (SDGs), which are an urgent call for action by all countries in a global partnership [26].

Scholars have completed studies relating to these goals from different fields and angles, and using different tools. For example, Sachs et al. [29] introduced six SDG transformations as modular building blocks of SDG achievement. Silvestre and Ţîrcăb [30] proposed a typology of innovation for sustainable development. Fuso Nerini et al. [31] suggest that climate change can undermine 16 SDGs, while combating climate change can reinforce all 17 SDGs but undermine efforts to achieve 12. Therefore, deeper study is needed to understand the relationship between climate change and sustainable development and to maximize the effectiveness of action in both domains. With the outbreak of COVID-19, Di Marco et al. [32] reminded that sustainable development must account for pandemic risk. Other studies have focused on sustainability strategy [33–35] or development [36–38] or different perspectives [39–42].

## Methodology

There are some researches focus on how COVID-19 affect the tourism industry. For example, Yeh [23] use a qualitative research method to examine the tourism crisis and disaster management. Fotiadis, Polyzos, and Huan [24] try to forecast the international tourist arrivals from July 2020 to June 2021. Kock et al. [25] investigating the COVID-19 effects on the tourists' psyche. So far, there are still no research using quantitative method to investigate the contribution of tourists from different country or purpose to Taiwan tourism industry.

Addressing the 8th, 9th, 12th, and 17th SDGs, the objective of this study is to find a sustainability strategy for Taiwan's tourism industry. Identifying the needs of international tourists and the niche of the industry, using existing resources to make a proper response, and using innovative concepts to ensure sustainable consumption and production methods will be part of this strategy. Promoting sustained inclusive and sustainable economic growth and revitalizing the global partnership for sustainable development will also be necessary. The research process of this study is described as Fig 1.

There are many techniques can be used for efficiency (contribution) evaluation. Some famous techniques are analytic hierarchy process (AHP) [43], balanced scorecard (BSC) [44], data envelopment analysis (DEA) [45], and Principal components analysis [46,47]. These techniques are easy to applied to real world situation and obtain useful results. However, AHP required some experts' subjective opinions to setup the weights for evaluation [43]. BSC need to consider four perspectives (Financial, Customer, Internal business processes, and Learning and growth) simultaneously [44].

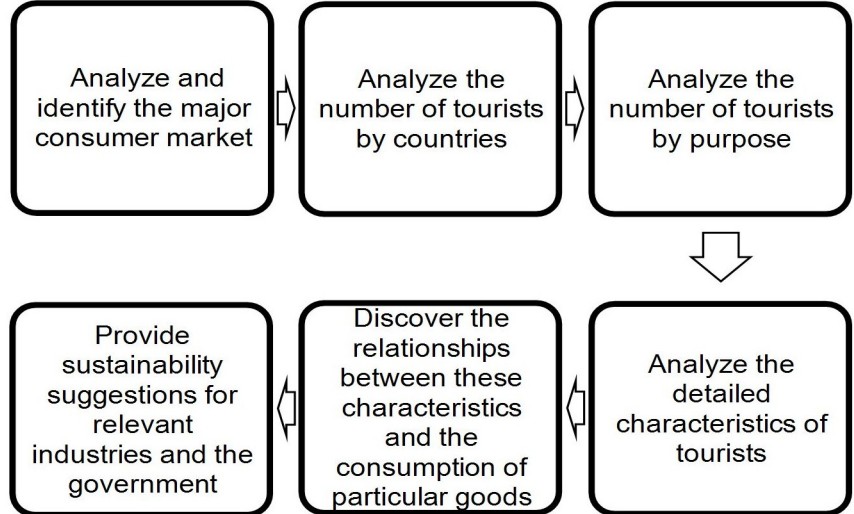

**Fig 1. Research process.**

In tourism industry, it is not suitable to let some experts decide which country or purpose is more important than the others are. Furthermore, this study analyze the existing data to find out the relative contribution of tourists from different country or purpose. Therefore, DEA and PCA techniques are more suitable than AHP and BSC techniques in this study.

## Two-stage DEA model

Charnes, Cooper, and Rhodes (CCR) [45] proposed the data envelopment analysis (DEA) to identify the relative efficiency of a group of decision-making units (DMUs). This evaluation methodology, commonly called the DEA-CCR model, has been widely employed in many fields and industries [48–51].

However, the DEA-CCR model can only be used to measure overall performance with initial inputs and outputs. Therefore, it cannot provide sufficient information for managers to assess the advantages and disadvantages of their competitive strategies if a corporation with a complicated system were to divide its performance into many kinds of segments and wants to measure the managerial abilities of each segment separately [52].

To overcome this shortage, a two-stage DEA model for providing more information of the inefficiency DMUs was introduced by [52]. In this model, a mediating factor was added to split the traditional DEA-CCR model into two stages; the mediating factor is the output of the first stage and the input of the second stage. The two-stage DEA can appropriately assess the managerial abilities of each segment and has been mentioned and applied in many fields. For example, Hwang and Kao [53] applied a two-stage DEA model to evaluate the managerial efficiency of non-life insurance companies. Gregoriou, Lusk, and Halperin [54] used a two-stage DEA model to measure the performance efficiency of U.S. national banks. Deyneli [55] used the two-stage DEA model to determine the relationship between the salaries of judges and the efficiency of justice service. Many researchers also applied the two-stage DEA model to evaluate relative efficiency in different fields [56–59]. Tsai et al. [60] pointed out that the two-stage DEA analysis can present more detailed information. Therefore, this study performs a two-stage DEA analysis to obtain suggestions for Taiwan's tourism industry in terms of which countries or aspects it should focus on to increase revenue.

Because the income levels of different countries and regions are different, it is not enough simply comparing total expenditure amounts without comparing detailed information on tourists from various countries. As mentioned above, a two-stage data envelopment analysis (DEA) can provide more information about the data [60]; therefore, this study uses the technique as a tool to analyze the expenditure of tourists from different countries or with different purposes to calculate the overall benefits to Taiwan's tourism industry.

In the first stage, the relative consumption, from different countries or regions and for different purposes, is analyzed. The money tourists spent on each item is compared to see how tourists from different countries or with different purposes contribute to Taiwan's tourism industry. To fit the requirements of the DEA model, the money tourists spent in each item is treated as an output factor, and the input factor is assumed to be the same and set as one unit. Results are shown in next section.

Note that in the DEA model, the computed efficiency value represents the relative output/input ratio of each DMU. A higher efficiency value means more output or less input. In this study, all output values are the consumption of tourists on each item, with a virtual input item set equal to 1. Therefore, the efficiency value can be treated as the contribution to the tourism industry, where a higher efficiency value means that the tourists spent more money. Such ideas can also wider the application of the DEA model.

## Principal component analysis

Principal component analysis (PCA) is a multivariate technique that analyzes a set of data in which observations can be described by several correlated variables [46,47]. When there are many data dimensions (variables), principal component analysis can reduce the number of dimensions (variables), but the data characteristics will not differ too much. It extracts the dominant patterns in the matrix. Typically, two or three principal components are usually sufficient [47]. According [61], PCA is a technique for reducing the dimensionality of such datasets, increasing interpretability but at the same time minimizing information loss, hence making PCA an adaptive data analysis technique.

PCA is the simplest method to analyze multivariate statistical distributions with feature quantities. Usually, this kind of operation can be seen as revealing the internal structure of the data, so as to better show the variability of the data. It has been applied in many fields. For example, Ni et al. [62] used PCA to evaluate civil aviation safety. Teng et al. [63] applied PCA for process improvement in industrial refineries. Wang et al. [64] used PCA in supply chain monitoring. Uddin et al. [65] applied PCA in mapping climate vulnerability in coastal regions of Bangladesh.

Because the spending amount of international tourists contains many items and subitems, to find the items that contributed most to Taiwan's tourism industry, this study also applies the PCA technique to reduce the number of variables.

## Results

### Analysis of major consumer markets

This study uses data from the Tourism Bureau [2,66] to collect details on amounts spent by tourists from different countries or regions in recent years to make a statistical analysis and comparison. Firstly, the major consumer markets can be identified from the inbound visitors' data. It can be seen from Table 2 that, in the past five years, tourists from Asia have grown the fastest, followed by the Americas. From Table 3, we see that tourists to Taiwan are mainly from China, Japan, Hong Kong (HK), Korea, the United States (USA), Malaysia, the Philippines (PH), and Singapore (SG). The number of tourists from China decreased sharply after

**Table 2. The number of tourists from each continent to Taiwan from 2015 to 2019.**

| Year | Asia | Americas | Europe | Oceania | Africa | Unknown |
|------|------|----------|--------|---------|--------|---------|
| 2015 | 3,766,249 | 641,957 | 353,112 | 110,492 | 10,572 | 665 |
| 2016 | 4,488,396 | 705,878 | 378,674 | 118,207 | 11,002 | 863 |
| 2017 | 5,144,314 | 757,025 | 410,805 | 127,770 | 12,006 | 1018 |
| 2018 | 5,484,556 | 783,560 | 425,814 | 138,687 | 12,054 | 1144 |
| 2019 | **6,068,826** | 818,847 | 464,231 | 152,326 | 12,862 | 1176 |

Source: [2].

**Table 3. The number of tourists from main countries or regions to Taiwan from 2015 to 2019.**

| Year | China | Japan | HK | Korea | USA | Malaysia | PH | SG |
|------|-------|-------|-----|-------|-----|----------|-----|-----|
| 2015 | 4,184,102 | 1,627,229 | 1,389,529 | 658,757 | 479,452 | 431,481 | 139,217 | 393,037 |
| 2016 | 3,511,734 | 1,895,702 | 1,474,521 | 884,397 | 523,888 | 474,420 | 172,475 | 407,267 |
| 2017 | 2,732,549 | 1,898,854 | 1,540,765 | 1,054,708 | 561,365 | 528,019 | 290,784 | 425,577 |
| 2018 | 2,695,615 | 1,969,151 | 1,506,536 | 1,019,441 | 580,072 | 526,129 | 419,105 | 427,222 |
| 2019 | **2,714,065** | **2,167,952** | **1,598,223** | **1,242,598** | 605,054 | 537,692 | 509,519 | 460,635 |

Source: [2].

the Democratic Progressive Party (DPP) came to power, but as of 2019, the number of Chinese tourists is still the largest.

From Table 4, we see that tourists to Taiwan are mainly 30–39 years old, followed by 20–29, 40–49, and 50–59. The number of tourists 20–39 years old grow rapidly in the past five years compared to the other age groups. Table 6 shows the numbers of tourists to Taiwan with different objectives. Leisure is the main purpose and grew rapidly over the five years. One interesting phenomenon is that tourists with "other" purposes kept increasing compared to business, visiting relatives, study, etc. and became the second-largest group. Government staff may need to add more survey items in the future so as to perform more in-depth research.

The first five lines in Table 6 show the average detailed consumption per person per day of tourists in Taiwan from 2015 to 2019. The major items are hotel expenses and shopping. Taiwan is a relatively small country, and food and beverage costs are cheaper than in neighboring countries. Therefore, it is not easy to increase catering and transportation (Trans.) income.

It is a pity that the money spent by international tourists on entertainment (Ent.) in Taiwan is quite low. This merits the government's and industries' efforts to promote the country's attractions and entertainment venues internationally so as to increase income. In addition, hoteliers should think about how to improve the quality of their services so that travelers are

**Table 4. The number of tourists of different ages to Taiwan from 2015 to 2019.**

| Year | 1–9 (Years) | 10–19 | 20–29 | 30–39 | 40–49 | 50–59 | 60 and Over |
|------|-------------|-------|-------|-------|-------|-------|-------------|
| 2015 | 366,358 | 575,074 | 2,014,944 | 2,126,664 | 1,885,288 | 1,778,173 | 1,693,284 |
| 2016 | 397,178 | 603,564 | 2,214,660 | 2,250,830 | 1,925,259 | 1,710,215 | 1,588,573 |
| 2017 | 414,241 | 623,659 | 2,267,311 | 2,335,200 | 1,917,526 | 1,663,481 | 1,518,183 |
| 2018 | 441,845 | 660,123 | 2,357,308 | 2,483,099 | 1,962,793 | 1,657,013 | 1,504,526 |
| 2019 | 482,046 | 735,662 | **2,444,593** | **2,622,115** | **2,090,279** | 1,793,508 | 1,695,902 |

Source: [2].

**Table 5. The number of tourists with different purposes to Taiwan from 2015 to 2019.**

| Year | Leisure | Business | Visit | Study | Conference | Medical | Exhibition | Other |
|------|---------|----------|-------|-------|------------|---------|------------|-------|
| 2015 | 7,505,457 | 758,889 | 408,034 | 59,204 | 60,777 | 67,298 | 13,749 | 1,566,377 |
| 2016 | 7,560,753 | 732,968 | 428,625 | 67,954 | 64,704 | 38,260 | 14,876 | 1,782,139 |
| 2017 | 7,648,509 | 744,402 | 455,429 | 73,135 | 66,519 | 30,764 | 16,274 | 1,704,569 |
| 2018 | 7,594,251 | 738,027 | 483,052 | 76,925 | 73,529 | 34,701 | 17,355 | 2,048,867 |
| 2019 | **8,444,024** | 746,115 | 478,220 | 80,630 | 76,308 | 55,937 | 18,320 | **1,964,551** |

Source: [2].

willing to spend more on accommodation. The government and other sales companies should also think about how to use their creativity to launch souvenirs and famous products that conform to the image of Taiwan to promote tourists' purchase intention.

Based on the information in Tables 2 to 5, it can be seen that, in the past five years, international tourists to Taiwan have mainly been young people (20–49 years old) from China, Japan, Korea, several Southeast Asian countries, and the United States. Furthermore, the main purpose is leisure. Especially after the government introduced the new southbound policy, the number of tourists from these countries has grown significantly. Therefore, this study focuses on these countries, discusses their consumption trends, and puts forward some specific suggestions with reference to relevant government units and industry operations.

## Two-stage DEA analysis

**First stage DEA analysis by country or region.** Using the data collected from the Tourism Bureau [66], the average consumption (per person per day) of tourists from different countries or regions from 2015 to 2019 was used to compare their relative contribution to Taiwan's tourism industry. The total money they spent and on each item are treated as an output factor in the DEA model. Using the DEA-Solver software (SAITECH: Holmdel, NJ, USA) to compute the relative score, this can be treated as their contribution to the industry. The data and results are summarized in Table 6. Note that the data of Singapore, Malaysia, New Zealand, Australia, and other Southeast Asian countries have been combined as the "New Southbound 18 Countries" since 2017, and the data from Hong Kong include Macau. Each year for each country or region is treated as one decision-making unit (DMU) in the DEA model.

It can be seen from Table 6 that, in terms of total consumption, travelers from Japan in these five years (2015–2019), travelers from China in 2015 and 2018, and travelers from Singapore in 2016 spent more in Taiwan. In terms of individual consumption, travelers from Japan spent more in those five years on hotel expenses (2015–2019), transportation expenses (2015–2016), and entertainment expenses (2015–2017). Travelers from China spent more money on shopping than other countries, especially in 2015 and 2018. Although their relative consumption in 2016, 2017, and 2019 was low, the amount of shopping in these three years was still higher than that of travelers from other countries or regions.

Another interesting thing is that, in 2018, tourists from Hong Kong and Macau spent the most money on food and beverage (dining) expenses. In 2018, tourists from the United States spent the most money on miscellaneous (Misc.) expenses. In 2016, tourists from Singapore spent the most money on hotel expenses. Tourists from Europe, the United States, Malaysia, New Zealand, and Australia have a relatively low total spending amount, rarely exceeding US $170 on average.

**First stage DEA analysis by purpose.** Table 7 shows the data and computed results of relative spending of international tourists with different purposes. Tourists who come to Taiwan

**Table 6. First stage DEA analysis results by country or region.**

| DMU | Total | Hotel | Shopping | Dining | Trans. | Ent. | Misc. | Score |
|---|---|---|---|---|---|---|---|---|
| All2015 | 207.87 | 67.02 | 72.10 | 32.77 | 27.62 | 6.49 | 1.87 | 0.908 |
| All2016 | 192.77 | 70.88 | 58.24 | 31.95 | 24.22 | 5.23 | 2.25 | 0.841 |
| All2017 | 179.45 | 67.47 | 50.81 | 34.04 | 18.07 | 5.58 | 3.48 | 0.813 |
| All2018 | 191.70 | 66.00 | 56.52 | 39.67 | 19.30 | 6.06 | 4.15 | 0.903 |
| All2019 | 195.91 | 76.62 | 51.74 | 38.48 | 18.75 | 6.03 | 4.29 | 0.892 |
| Japan2015 | **227.59** | 97.32 | 41.69 | 39.06 | **35.89** | **11.81** | 1.82 | **1** |
| Japan2016 | **241.42** | 108.73 | 43.18 | 41.99 | **33.44** | **10.89** | 3.19 | **1** |
| Japan2017 | **214.05** | 101.85 | 40.68 | 37.68 | 18.69 | 10.08 | 5.07 | 0.981 |
| Japan2018 | **219.35** | 104.21 | 39.13 | 41.63 | 19.24 | 9.39 | 5.75 | 0.955 |
| Japan2019 | **229.42** | **110.40** | 39.16 | 44.24 | 19.36 | 9.92 | 6.34 | **1** |
| Korea2015 | 207.78 | 92.20 | 44.38 | 35.68 | 28.19 | 4.66 | 2.67 | 0.890 |
| Korea2016 | 188.06 | 80.93 | 41.71 | 37.71 | 23.36 | 2.68 | 1.67 | 0.854 |
| Korea2017 | 194.58 | 77.10 | 46.29 | 41.19 | 18.44 | 7.05 | 4.51 | 0.899 |
| Korea2018 | 187.71 | 73.03 | 39.67 | 45.42 | 18.23 | 6.30 | 5.06 | 0.961 |
| Korea2019 | 201.96 | 97.58 | 35.78 | 40.52 | 17.73 | 4.49 | 5.86 | 0.914 |
| China2015 | **227.58** | 43.67 | **120.03** | 27.22 | 29.45 | 5.62 | 1.59 | **1** |
| China2016 | 198.43 | 45.50 | 96.30 | 25.22 | 25.07 | 4.52 | 1.82 | 0.875 |
| China2017 | 184.38 | 49.69 | 83.08 | 27.73 | 16.32 | 4.75 | 2.81 | 0.851 |
| China2018 | **211.68** | 50.44 | **105.31** | 30.37 | 16.50 | 5.33 | 3.73 | **1** |
| China2019 | 199.63 | 53.55 | 91.26 | 29.36 | 15.67 | 5.81 | 3.98 | 0.961 |
| HongKong2015 | 184.76 | 69.49 | 50.70 | 37.11 | 20.48 | 4.93 | 2.05 | 0.851 |
| HongKong2016 | 182.98 | 71.30 | 49.09 | 35.67 | 20.01 | 4.30 | 2.61 | 0.835 |
| HongKong2017 | 183.92 | 63.93 | 51.30 | 41.55 | 19.64 | 4.70 | 2.80 | 0.881 |
| HongKong2018 | 202.31 | 61.94 | 60.65 | **49.55** | 20.90 | 5.38 | 3.89 | **1** |
| HongKong2019 | 208.58 | 79.55 | 52.47 | 46.28 | 20.61 | 5.62 | 4.05 | 0.985 |
| USA2015 | 163.62 | 82.00 | 21.35 | 33.75 | 17.90 | 5.93 | 2.69 | 0.768 |
| USA2016 | 149.03 | 71.65 | 22.84 | 27.35 | 15.79 | 4.83 | 6.57 | 0.785 |
| USA2017 | 155.67 | 75.53 | 17.75 | 35.51 | 17.84 | 3.77 | 5.27 | 0.809 |
| USA2018 | 159.42 | 62.00 | 20.88 | 40.33 | 18.76 | 5.41 | **12.04** | **1** |
| USA2019 | 171.61 | 73.24 | 28.94 | 40.53 | 18.71 | 5.39 | 4.80 | 0.879 |
| Europe2015 | 158.06 | 94.30 | 13.27 | 26.53 | 16.00 | 5.48 | 2.48 | 0.848 |
| Europe2016 | 132.07 | 70.74 | 13.62 | 25.02 | 16.34 | 3.23 | 3.12 | 0.644 |
| Europe2017 | 137.19 | 75.07 | 11.09 | 27.37 | 19.04 | 2.88 | 1.74 | 0.680 |
| Europe2018 | 148.15 | 74.89 | 11.47 | 36.19 | 18.94 | 3.11 | 3.55 | 0.798 |
| Europe2019 | 146.68 | 75.29 | 13.13 | 32.09 | 18.95 | 5.22 | 2.00 | 0.730 |
| Singapore2015 | 205.08 | 81.14 | 51.50 | 40.66 | 25.24 | 4.75 | 1.79 | 0.932 |
| Singapore2016 | **229.43** | **111.46** | 45.81 | 40.22 | 24.60 | 5.21 | 2.13 | **1** |
| Malaysia2015 | 162.07 | 54.07 | 43.52 | 29.40 | 26.15 | 6.35 | 2.58 | 0.806 |
| Malaysia2016 | 142.45 | 54.53 | 39.63 | 23.69 | 18.97 | 2.77 | 2.86 | 0.654 |
| NewZ&Aust2015 | 161.57 | 74.67 | 35.35 | 29.06 | 15.25 | 5.85 | 1.39 | 0.714 |
| NewZ&Aust2016 | 142.35 | 78.89 | 16.27 | 22.76 | 17.35 | 4.39 | 2.69 | 0.713 |
| New18South2017 | 152.25 | 55.36 | 42.43 | 29.64 | 17.63 | 3.89 | 3.30 | 0.711 |
| New18South2018 | 165.81 | 51.22 | 47.04 | 37.15 | 20.82 | 5.36 | 4.22 | 0.850 |
| New18South2019 | 170.46 | 63.77 | 41.63 | 36.31 | 19.98 | 5.50 | 3.27 | 0.792 |

Source: [66].

**Table 7. First stage DEA analysis results by purpose.**

| DMU | Total | Hotel | Shopping | Dining | Trans. | Ent. | Misc. | Score |
|---|---|---|---|---|---|---|---|---|
| Leisure2015 | 214.04 | 59.89 | 83.77 | 32.37 | **29.69** | 6.64 | 1.68 | **1** |
| Leisure2016 | 197.65 | 66.69 | 66.14 | 31.46 | 25.99 | 5.55 | 1.82 | 0.914 |
| Leisure2017 | 185.44 | 65.53 | 58.59 | 33.57 | 18.08 | 6.13 | 3.54 | 0.852 |
| Leisure2018 | 200.32 | 65.15 | *64.75* | *40.33* | *19.48* | *6.57* | 4.04 | **1** |
| Leisure2019 | 203.55 | 78.12 | 56.62 | 38.94 | 18.83 | 6.77 | 4.27 | 0.980 |
| Business2015 | 232.80 | **128.82** | 34.01 | 37.28 | 24.62 | 5.86 | 2.21 | **1** |
| Business2016 | 220.21 | 126.67 | 29.94 | 35.61 | 22.41 | 3.53 | 2.05 | 0.983 |
| Business2017 | 215.92 | 119.97 | 24.79 | 40.31 | 24.23 | 4.17 | 2.45 | 0.998 |
| Business2018 | 216.53 | *117.21* | 26.21 | *41.34* | *25.09* | 4.01 | 2.67 | **1** |
| Business2019 | 222.48 | *123.22* | 28.81 | *39.71* | *25.05* | 3.87 | 1.82 | **1** |
| Exhibit2015 | *267.93* | *77.46* | *50.34* | 30.83 | *24.91* | *9.53* | *74.86* | **1** |
| Exhibit2016 | 212.77 | *101.56* | 35.04 | *34.08* | *27.87* | *10.18* | 4.04 | **1** |
| Exhibit2017 | 201.88 | 91.41 | 46.36 | 36.64 | 17.46 | 3.21 | 6.80 | 0.889 |
| Exhibit2018 | *257.21* | *101.93* | *79.32* | *38.53* | *23.17* | *7.75* | 6.51 | **1** |
| Exhibit2019 | 181.98 | 80.73 | 41.07 | 28.48 | 18.11 | 2.34 | 11.25 | 0.742 |
| Visit2015 | 125.85 | 42.31 | 31.75 | 30.20 | 12.82 | 6.14 | 2.63 | 0.786 |
| Visit2016 | 123.79 | 35.29 | 37.25 | 30.50 | 12.04 | 4.37 | 4.34 | 0.751 |
| Visit2017 | 108.09 | 27.42 | 32.03 | 29.33 | 11.89 | 3.73 | 3.69 | 0.718 |
| Visit2018 | 117.90 | 27.33 | 34.13 | 32.97 | 12.64 | 3.77 | 7.06 | 0.803 |
| Visit2019 | 117.20 | 29.53 | 33.73 | 32.89 | 12.94 | 3.83 | 4.28 | 0.801 |
| Study2015 | 84.14 | 34.89 | 12.74 | 16.47 | 10.16 | 4.03 | 5.85 | 0.453 |
| Study2016 | 101.74 | 34.80 | 15.17 | 14.68 | 11.06 | 3.09 | 22.94 | 0.413 |
| Study2017 | 87.13 | 33.27 | 13.99 | 18.15 | 10.49 | 3.83 | 7.40 | 0.479 |
| Study2018 | 124.71 | 33.22 | 15.71 | 23.07 | 14.41 | **12.10** | 26.20 | **1** |
| Study2019 | 121.38 | 41.83 | 19.68 | 19.09 | 7.18 | 1.66 | 31.94 | 0.460 |
| Medical2015 | 336.83 | 96.18 | 80.79 | 31.44 | 20.61 | 1.23 | 106.58 | 0.978 |
| Medical2016 | *540.29* | *78.02* | *67.17* | *42.09* | *19.65* | 0.70 | *332.66* | **1** |
| Medical2017 | 526.94 | 67.20 | **177.72** | 40.00 | 11.76 | 2.12 | 228.14 | **1** |
| Medical2018 | 382.51 | 41.11 | 71.34 | **42.11** | 17.83 | 2.03 | 208.09 | **1** |
| Medical2019 | **856.92** | 63.49 | 62.73 | 38.46 | 15.55 | 2.14 | **674.55** | **1** |

Source: [66].

for medical purposes had the highest total spending amount in these five years, especially in 2016 and 2019. They spent most money on shopping in 2017 and on dining in 2018. They also spent more than average on hotel, shopping, dining, transportation, miscellaneous, and total amount in 2016. Visitors to exhibitions spent more than average on many items (mark in italics) in 2015, 2016, and 2018. Visitors for business also spent more than average on hotel, dining, and transportation in 2018, and 2019. They spent the most on hotel expense in 2015. Visitors for leisure purpose spent more than average on shopping, dining, transportation, and entertainment in 2018 and they spent the most on transportation in 2015.

Those who visit relatives or study (except in 2018, when they spent the most money in entertainment) have the lowest spending amount. Visitors who participate in exhibitions and business have a relatively high average spending on hotels. Travelers for leisure purposes spend more money on shopping. Travelers who are in Taiwan for medical purposes spend more on miscellaneous expenses; these may include their medical expenses.

**Table 8. Second stage DEA analysis results by details of consumption.**

| DMU | S1 | S2 | S3 | S4 | S5 | S6 | S7 | S8 | S9 | S10 | Score |
|---|---|---|---|---|---|---|---|---|---|---|---|
| All15 | 40.45 | 24.01 | 7.28 | 13.38 | 36.48 | 5.25 | 2.39 | 1.39 | 7.72 | 1.03 | 0.952 |
| All16 | 8.38 | 36.66 | 9.13 | 14.12 | 38.87 | 3.76 | 2.02 | 0.92 | 6.54 | 1.29 | 0.896 |
| All17 | 7.99 | 22.24 | 7.05 | 11.06 | 34.37 | 3.52 | 3.08 | 0.30 | 6.79 | 1.60 | 0.775 |
| All18 | 9.06 | 28.72 | 6.38 | 15.61 | 36.23 | 4.14 | 4.01 | 0.67 | 7.98 | 2.08 | 0.850 |
| All19 | 8.70 | 25.10 | 7.75 | 12.28 | 36.65 | 2.91 | 4.85 | 0.43 | 9.21 | 1.81 | 0.795 |
| Jap15 | 16.57 | 2.01 | 2.97 | 1.91 | *38.70* | 3.06 | 0.28 | 0.12 | 7.33 | 0.45 | 0.458 |
| Jap16 | 3.73 | 10.10 | 3.43 | 1.84 | *48.72* | 1.58 | 1.01 | 0.29 | *12.00* | 0.73 | 0.490 |
| Jap17 | 6.10 | 6.85 | 5.05 | 2.18 | *37.18* | 1.42 | 1.39 | 0.56 | *10.65* | 2.70 | 0.491 |
| Jap18 | 1.75 | 3.33 | 3.42 | 1.61 | *38.98* | 0.93 | 0.24 | 0.08 | *13.67* | 0.08 | 0.415 |
| Jap19 | 2.04 | 8.38 | 3.83 | 1.95 | *38.86* | 0.66 | 1.30 | 0.01 | *19.18* | 0.90 | 0.477 |
| Chn15 | **47.58** | 30.75 | 7.74 | 16.57 | 34.79 | 6.27 | 2.53 | 1.81 | 8.17 | 1.21 | **0.982** |
| Chn16 | 9.46 | **48.46** | 9.93 | 18.28 | 34.98 | 4.67 | 2.21 | 1.25 | 5.47 | 1.45 | **0.974** |
| Chn17 | 8.26 | 40.20 | 5.84 | 18.09 | 30.95 | 4.31 | 4.64 | 0.51 | 5.96 | 2.57 | 0.934 |
| Chn18 | 11.44 | 47.91 | 5.37 | **24.12** | 35.38 | 6.21 | 4.33 | 1.01 | 7.84 | 5.55 | **1** |
| Chn19 | 10.96 | 36.94 | 9.13 | 17.86 | 34.04 | 4.04 | 5.93 | 0.67 | 7.35 | 3.09 | 0.924 |
| Kor17 | 6.70 | 9.03 | 13.47 | 5.84 | 32.56 | 6.92 | 1.87 | 0.14 | 1.12 | 0.97 | 0.573 |
| Kor18 | 1.60 | 3.81 | 16.50 | 4.79 | 35.45 | 4.18 | 0.92 | 0.23 | 3.61 | 1.64 | 0.550 |
| Kor19 | 0.43 | 2.61 | 5.53 | 1.41 | *41.29* | 3.07 | 0.07 | 0.00 | 3.22 | 1.27 | 0.414 |
| New17 | 11.60 | 3.47 | 6.82 | 8.82 | 26.83 | 1.43 | 3.31 | 0.13 | 3.33 | 11.37 | 0.719 |
| New18 | 12.74 | 7.87 | 7.23 | 7.82 | 27.16 | 0.15 | 10.60 | 0.25 | 4.56 | 0.60 | 0.676 |
| New19 | 12.97 | 3.30 | 7.57 | 6.38 | 34.73 | 0.38 | 5.56 | 0.07 | 6.37 | 0.52 | 0.648 |

Source: [66].

It would be great if we could expand the total number of tourists. From another perspective, if we focus on attracting international tourists to Taiwan for medical treatment, exhibitions, business, and leisure, we can attract considerable income to Taiwan's tourism and medical industries. In particular, we might focus on medical tourism, which is becoming common and combines medical treatment and the use of different industries. In combination with other industries, it will generate considerable benefits.

**Second stage DEA analysis by detailed consumption.** In the second stage, the relative spending on each subitem by tourists from different countries or regions is analyzed. The money tourists spent on each subitem is compared to see how tourists from different countries contribute to Taiwan's tourism industry. In order to obtain an in-depth understanding of the shopping done by tourists when visiting Taiwan, this study uses data from the Tourism Bureau [66] to analyze the consumption details of tourists from major countries in the past five years (2015–2019). Collected data are listed in Table 8. The details on money spent by tourists from Japan (Jap) and China (Chn) include all five years. However, the Tourism Bureau start collecting data of Korea (Kor) and the new southbound 18 countries (New) in 2017, so only three years of data are presented. "All" represents the average purchase amount of tourists from all countries for sightseeing purposes.

In Table 8, S1 represents "Clothing or related accessories," S2 represents "Jewelry or jade," S3 represents "Souvenirs or handicrafts," S4 represents "Cosmetics or perfume," S5 represents "Famous product or specialty," S6 represents "Tobacco or alcohol," S7 represents "Chinese medicine or health food," S8 represents "Electronics or electrical appliances," S9 represents

""tea"," and S10 represent "others." These data are set as the output items in the second stage of the DEA.

To assess the contribution to the tourism industry of each DMU in detail, the total spending amount of each DMU (which is directly related to the details on consumption) in Table 7 is set as the input items in the second stage of the DEA. The score can represent the amount of money spent on different subitems by tourists from different countries in each year as the contribution to Taiwan's tourism industry. Using the DEA-Solver software to compute the relative score, the data and results are summarized in Table 8. Because the number of DMUs is relatively small compared to the input and output items, in order to improve the degree of discrimination, when calculating the efficiency by DEA-Solver, the weight of each input/output items value is set strictly.

From Table 8, all tourists spent most money on "Famous product or specialty", follow by "Jewelry or jade". In 2015, the average spending amount on "Clothing or related accessories" reach 40.45 USD. This may be affected by the consumption of China tourists. Tourists from China contributed the most in terms of shopping; they spent the most money on "Jewelry or jade" followed by "Famous product or specialty". Generally speaking, tourists from China spent no less than the average of all tourists on most of the subitems, except for "Souvenirs or handicrafts" in 2017 and 2018, "Famous product or specialty" in 2015–2019, and "tea" in 2016–2019, even though the total amount of money they spent was less than the average level.

From Table 8, we see that tourists from China spent most money on "Clothing or related accessories" in 2015; they spent most on "Jewelry or jade" in 2016; and they spent most on "Cosmetics or perfume" in 2018. In comparison, they spent less in total (input), which makes their consumption on these subitems higher than average. Therefore, the relative efficiency score was 1 in 2018 and greater than 0.9 in other years. Intuitively, we know that tourists from China contributed a lot to Taiwan's tourism industry in the "Shopping" item, even though their total spends was less than the average consumption of all countries.

Tourists from Japan spent most of their money on "Famous product or specialty" followed by "tea". Generally, Japanese tourists spent a lot more money than the average on these two subitems. However, compared to the total amount they spent in those years, the relative contribution was less than 0.5.

On the other hand, tourists from Korea prefer "Tobacco or alcohol". They spent relatively more money on these in 2017–2019 than average. In 2017 and 2018, they spent more on "Souvenirs or handicrafts"; their preference switch to "Famous product or specialty" in 2019. Tourists from Korea spent less than average on all other subitems. Tourists from New Southbound countries spent the most money on other items in 2017. They spent more than average on "Clothing or related accessories" and "Chinese medicine or health food" in 2017–2019. Generally, tourists from New Southbound countries spent a lot less than the average on other subitems.

One thing that needs to be noted is that the total money tourists spent on "Shopping" has continued to decrease slightly. The industry must increase the competitiveness of its products. After all, travelers can choose to go anywhere in the world. If they can easily find better similar products in different countries, they will not spend their money in Taiwan.

### Principal component analysis by item and subitem

To further understand the relationships between the items and subitems on which tourists spent their money, a principal component analysis (PCA) was applied to the amount spent on each item and subitem. First, a PCA was performed on tourists from different countries or regions, and two main components were extracted. The PCA results are shown in Fig 2. The

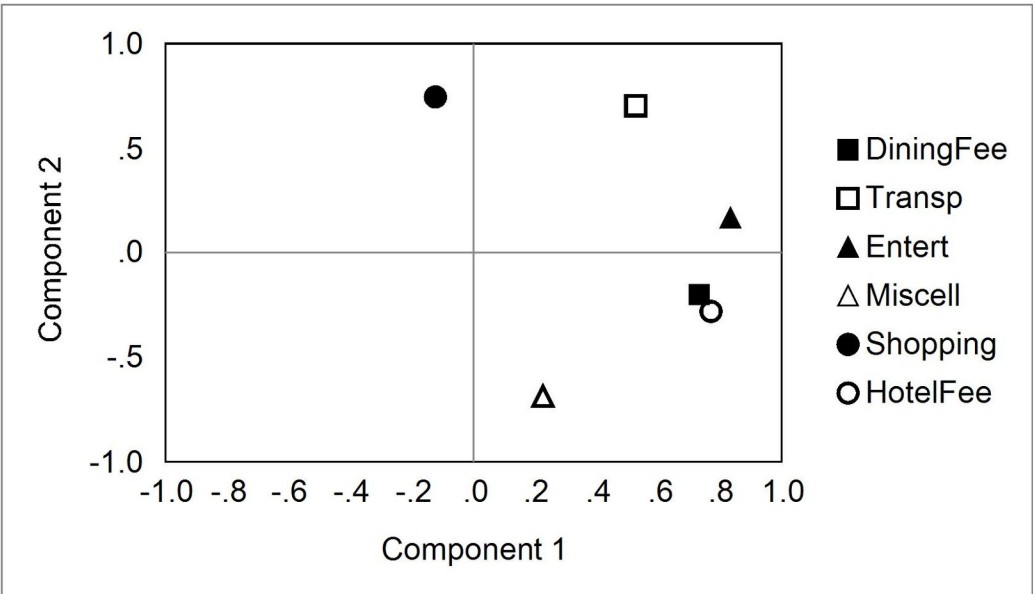

**Fig 2. The PCA analysis of money spent, by country.**

first component was hotel expenses, dining expenses, and entertainment expenses. The factor loadings were 0.770, 0.732, and 0.833, respectively. The second component was shopping expenses, transportation expenses, and miscellaneous expenses. The factor loadings were 0.745, 0.703, and –0.687, respectively. There was a high and significantly positive relationship between hotel expenses, entertainment expenses, and dining expenses. Travelers who spent more on hotel expenses also spent more on entertainment expenses and dining expenses. On the other hand, travelers who spent more on shopping and transportation had lower spending on miscellaneous expenses.

It can be seen from Fig 2 that if a travel agent makes itinerary plans for tourists from different countries, tourists from countries that are willing to spend more on accommodation are also willing to spend more on catering and entertainment, and vice versa. On the other hand, travelers who are willing to spend more on shopping generally also spend more on transportation, but less on miscellaneous expenses.

Secondly, a PCA was performed on tourists who visit Taiwan for different purposes, and two main components were extracted. The PCA results are shown in Fig 3. The first component was transportation expenses, hotel expenses, and dining expenses. The factor loadings were 0.916, 0.863, and 0.787, respectively. The second component was miscellaneous expenses, shopping expenses, and entertainment expenses. The factor loadings were 0.820, 0.675, and -0.680, respectively. There was a high and significantly positive relationship between hotel expenses, dining expenses, and transportation expenses. Travelers who spent more on hotel expenses also spent more on dining and transportation. On the other hand, travelers who spent more on shopping also spent more on miscellaneous expenses, but spent less on entertainment expenses.

It can be seen from Fig 2 that if the travel agent makes itinerary plans for tourists with different purposes, tourists who are willing to spend more on accommodation are also willing to spend more on catering and transportation, and vice versa. On the other hand, travelers who are willing to spend more on shopping generally also spend more on miscellaneous expenses, but less on entertainment.

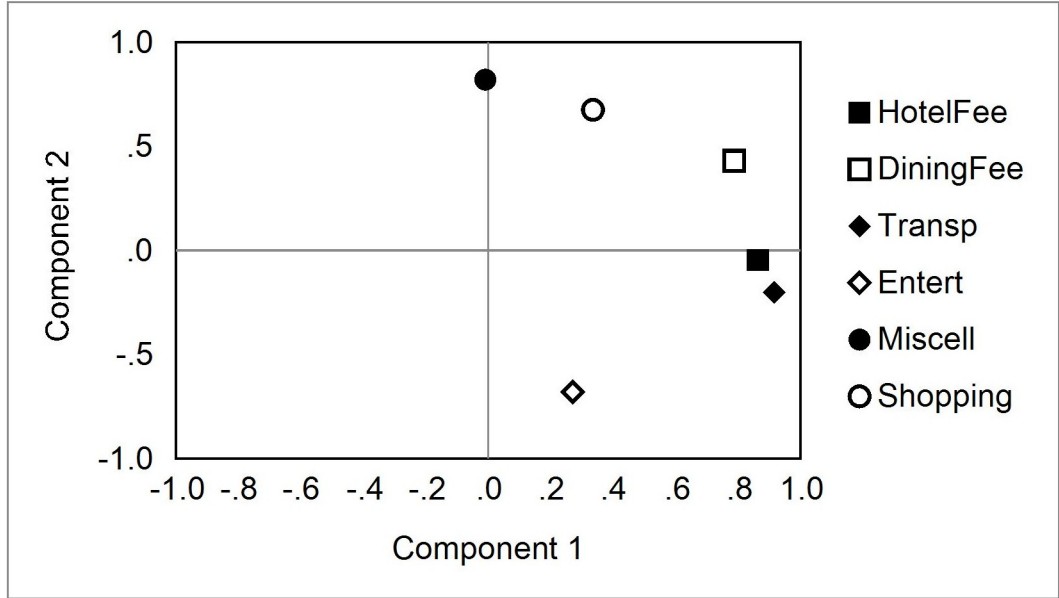

**Fig 3. The PCA analysis of money spent, by purpose.**

Finally, a PCA was performed on the items tourists spent money on, and two main components were extracted. The PCA results are shown in Fig 4. The first component included "Clothing or related accessories," "Jewelry or jade," "Cosmetics or perfume," "Tobacco or alcohol," and "Electronic or electrical appliances". The factor loadings were 0.659, 0.859, 0.843, 0.774, and 0.943, respectively. The second component included "Souvenirs or handicrafts," "Famous product or specialty," "Chinese medicine or health food," "tea," and "others." The factor loadings were 0.564, –0.884, 0.625, -0.758, and 0.532, respectively.

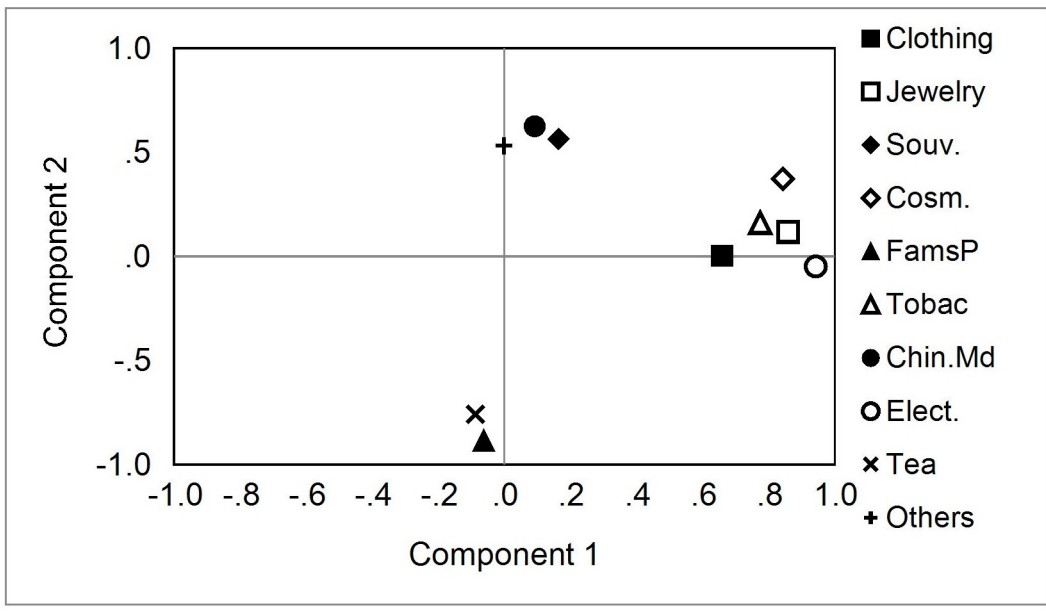

**Fig 4. The PCA analysis of money spent on particular items.**

There is a high and significant positive relationship among "Jewelry or jade," "Tobacco or alcohol," "Electronic or electrical appliances," "Cosmetics or perfume," and "Clothing or related accessories." Travelers who spend more on any one subitem also spend more on the other four subitems. On the other hand, travelers who spent more on "Famous product or specialty" or "tea" had lower spending on "Souvenirs or handicrafts," "others," and "Chinese medicine or health food."

The results in Fig 4 can alert the relevant industries that certain products are suitable for selling as a set, such as "Jewelry or jade," "Cosmetics or perfume," "Tobacco or alcohol," "Clothing or related accessories," and "Electronics or electrical appliances," which would be convenient for tourists to buy and would thereby increase sales.

## Discussion

From the analysis of the results in the previous section, it can be seen that the major tourists to Taiwan are from Asia and the Americas (Table 2), and the numbers of tourists from Southeast and Northeast Asia are growing particularly rapidly (Table 3). In terms of the number of people, the 30–39-year-old age group is the largest, followed by 20–29, 40–49, 50–59, and 60 or above (Table 4). Most of the tourists visit Taiwan for leisure, while the second-largest group is "others" (Table 5).

Unsurprisingly, travelers spend the most on hotel expenses and shopping (Table 6). Japanese tourists have the most spending, followed by Chinese and Singaporean tourists. It is worth noting that the number of tourists for medical purposes is not large, but their total consumption amount is the highest (Table 7). The Taiwanese government has done a very good job of containing the COVID-19 epidemic, so there are even surplus medical supplies that can help other countries. This is a good publicity opportunity. If the government and related industries take advantage of the good reputation the country has for its handling of coronavirus, we should be able to increase the number of tourists coming to Taiwan for medical treatment, which should be of great help to the overall medical and tourism industry.

In terms of shopping (Table 8), tourists from all countries spent at least US$34 on "Famous product or specialty." Chinese tourists spent the most money on "Jewelry or jade," which was also a favorite subitem for all tourists; they spent more than US$20 on it on average. However, tourists from Japan, Korea, and the New Southbound countries spent less than average on this subitem. "Cosmetics or perfume," "Souvenirs or handicrafts," "tea," and "Clothing or related accessories" also attracted tourists' interest. Especially in 2015, the average amount tourists spent on clothes reached US$40.45. Relevant units can investigate and see if this particularly high amount is due to the influence of Chinese tourists, or of tourists from other countries who also like to buy this subitem. The industry can try to increase its product advantages to attract international tourists. After all, tourists who revisit the same place may not want to buy the same product twice.

According to the Sustainable Development Goals, a sustainability strategy of operation should make good use of existing resources to get the most rewards and should be able to operate sustainably. Therefore, without wasting resources or doing unnecessary construction or investment, we should analyze the preferences of existing and potential consumers to take better advantage of the existing tourism environment. In order to attract international tourists to Taiwan, we need to provide better service quality. The pursuit of growth that is beneficial to both industries and tourists is the goal of this study. The results of the principal component analysis show that travelers have some particular norms to their shopping behaviors, in that the items and subitems they bought show some correlations (Figs 2–4). From the above

analysis, we see that this study provides several suggestions for relevant government units and industry operators.

1. Focus advertising on foreign tourists, mainly on young and middle-aged customers (20–49 years old), because they have higher spending power and autonomy.

2. The Tourism Bureau must add more items to their investigation of tourists with the purpose of "others," so as to understand in more detail their reasons for visiting in the future.

3. From the data envelopment analysis (DEA) results, for Japanese, Chinese, and Singapore travelers with high spending power, we must develop more options, and produce souvenirs with local characteristics to increase their purchase intention. Aiming at Chinese tourists, we should develop more high-quality, high-unit-price products in the areas of "Jewelry or jade," "Famous product or specialty," "Clothing or related accessories," and "Cosmetics or perfume" with Taiwanese characteristics to increase their purchase intention.

4. Dealing with the coronavirus pandemic will involve cooperating with other countries and letting people know about Taiwan's medical environment and successful cases. This could increase the willingness of foreign tourists to come to Taiwan for medical treatment.

5. Tourists in Southeast Asia have different dietary requirements from those of ordinary people in Taiwan. Relevant industries must focus more on their needs so that these tourists feel more at ease and willing to travel to Taiwan.

6. From the principal component analysis (PCA) results, operators in different industries can make different and diversified combinations of their products, so that passengers can buy products that are more valuable for less money, thereby increasing their purchase intention.

## Conclusions

The outbreak of a new coronavirus (COVID-19) has caused great damage to the global economy. The tourism industry is among the worst-hit industries. Currently, many countries are focusing on developing a vaccine to control the epidemic. I hope that the pandemic may be contained in the near future, but we do not know if global consumer behavior remain the same. Therefore, how to focus on the passengers who are most helpful to Taiwan's tourism industry is a very important question.

So far, most of the research related to COVID-19 has focused on patients' symptoms, transmission, treatment and prevention, etc. [9–16]. Some are focus on the political and economic impact [17–22] or simply the impact to tourism industry [23–25]. With "The 2030 Agenda for Sustainable Development," this study tries to find a sustainability strategy for Taiwan's tourism industry. Using two-stage data envelopment analysis and principal component analysis, this study investigates past statistics and explores the behavior of tourists who travel to Taiwan. Specific recommendations for a sustainability strategy are made with reference to relevant industries, especially for tourists from China, Japan, and Southeast Asian countries. As mentioned in previous section.

This study not only discusses the economic impact of COVID-19 on Taiwan's tourism industry, but also tries to find a way forward for Taiwan's tourism industry from a sustainable development perspective. The bottom line is this: Do not overinvest; make good use of existing resources to maintain a competitive strategy.

From the perspective of the global tourism market, Taiwan's tourism industry still has considerable room for growth. Under the sustainable development goals of the United Nations [26], how to use resources effectively without increasing waste and pursuing sustainable

management is a hugely important question [29–33,36–39]. This research proposes some suggestions by investigating shopping behavior. In theory, to make the application of the DEA model broader, subsequent researchers can expand this concept. In practical applications, combining the concepts of DEA and PCA can be an effective way to determine tourists' consumption patterns. Such a finding can help related businesses to develop their products, find ways to cooperate with each other, and increase their sales. However, follow-up researchers can still do more in-depth research on how to operate sustainably from different perspectives.

## Acknowledgments

I appreciate the multiple anonymous reviewers for identifying the shortcomings of this article and providing insights and suggestions that contributed greatly to its improvement.

## Author Contributions

**Conceptualization:** Ming-Chi Tsai.

**Data curation:** Ming-Chi Tsai.

**Formal analysis:** Ming-Chi Tsai.

**Investigation:** Ming-Chi Tsai.

**Methodology:** Ming-Chi Tsai.

**Project administration:** Ming-Chi Tsai.

**Resources:** Ming-Chi Tsai.

**Software:** Ming-Chi Tsai.

**Supervision:** Ming-Chi Tsai.

**Validation:** Ming-Chi Tsai.

**Visualization:** Ming-Chi Tsai.

**Writing – original draft:** Ming-Chi Tsai.

**Writing – review & editing:** Ming-Chi Tsai.

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
