## [Decision Letter · Decision Letter 0]

23 Nov 2020

PONE-D-20-33157

Developing a sustainability strategy for Taiwan’s tourism industry after the COVID-19 pandemic

PLOS ONE

Dear Dr. Tsai,

Thank you for submitting your manuscript to PLOS ONE. After careful consideration, we feel that it has merit but does not fully meet PLOS ONE’s publication criteria as it currently stands. Therefore, we invite you to submit a revised version of the manuscript that addresses the points raised during the review process.

We look forward to receiving your revised manuscript.

Kind regards,

Bing Xue, Ph.D.

Academic Editor

PLOS ONE

Journal Requirements:

Reviewers' comments:

Reviewer's Responses to Questions

**Comments to the Author**

1. Is the manuscript technically sound, and do the data support the conclusions?

Reviewer #1: Yes

Reviewer #2: Partly

2. Has the statistical analysis been performed appropriately and rigorously? 

Reviewer #1: Yes

Reviewer #2: I Don't Know

3. Have the authors made all data underlying the findings in their manuscript fully available?

Reviewer #1: Yes

Reviewer #2: No

4. Is the manuscript presented in an intelligible fashion and written in standard English?

Reviewer #1: Yes

Reviewer #2: No

5. Review Comments to the Author

Reviewer #1: Dear Editor and Author(s),

I read the manuscript and I found it very beneficial manuscript to serve the country in best regarding with the tourism. The paper would provide the items can be sold to the tourist in their country with the best demanded one. Therefore, to help the country regarding with the products and services in best this paper would serve in best.

Reviewer #2: Introduction:

Firstly, why do you use two-stage data envelopment analysis (DEA) and principal component analysis (PCA)? Are there any obvious benefit of DEA and PCA in this study, and what are the drawbacks of other major methods in this study? Or put why you use these methods in the "Methodology part" for explanation.

Secondly, the application value of the paper is clearly explained, but does this study have theoretical and academic value? Whether there are academic theoretical shortcomings that need to be remedied in the context of the Taiwan region regarding the economic recovery of tourism after the epidemic? The academic research question of the paper is not very clear? Are you trying to find out where is the highest consumption area in Taiwan, or the greatest contribution to the tourism industry in Taiwan? From my judgment, you may want to look for reasonable sustainability strategy for the tourism economy after the COVID-19, and those small questions ( the highest consumption area or the greatest contribution to the tourism industry) are only for this main research question, therefore, in the Introduction part, you need to concentrate on deriving your research question reasonably and explaining its theoretical significance (if there is an ideal theoretical significance), so that it seems to be more logical in the Introduction part, otherwise it may make readers confused.

Related Works:

firstly, delete “The United Nations World Tourism Organization” (in line 83), because you've explained in line 56 exactly what UNWTO stands for.

Secondly, the part (the Current status of the international tourism market) is actually quite repetitive with the Introduction part, so it is suggested to simplify this part or enrich the main content of this part into the Introduction part.

Thirdly, “New vaccines may be developed soon, so there is hope that the epidemic will end within a few months. However, international tourists may not come back as soon as expected. Moreover, consumer behavior may change temporarily or even forever. Therefore, how to attract international tourists to visit Taiwan again as soon as possible is a very important question, and the country must focus on those tourists who are most helpful to their operational performance so that the industry can recover in the shortest time possible.” (in line 137-141)

Whether these above contents have basis or document source, or just your guess and judgment.

Methodology:

It is recommended to make a table or figure to illustrate the benefits of the two methods (in fact, you had mentioned in your literature review, but they look scattered.), and point out why it is suitable to analyze statistics of visitors to Taiwan from the Tourism Bureau, so that the characteristics of the research methods ( DEA-CCR model and PCA) will be more clear.

Results:

Firstly, the part of “Two-stage DEA analysis” (in line 247-263) is recommended to be put into the "Methodology" part, because it is redundant to put it in the "Results" part.

Secondly, you should know that COVID-19 is probably confirmed by outstanding medical scientists around the world and may have existed before December 2019, but COVID-19 really obviously negative affects the global tourism industry after January 2020. and it is possible that the Taiwan region will be affected by the obvious negative impact of the new crown epidemic after January 2020. And you used the DEM method to analyze the data from 2014 to 2018. Although it can reflect the distribution and consumption of tourists to Taiwan during the period before the new crown epidemic to a certain extent, there is still a lack of key tourist statistics for 2019. You used the Two-stage DEA method to analyze the data from 2014 to 2018, although it can reflect to a certain extent the distribution and consumption of tourists who visited Taiwan during the period before the COVID-19 pandemic, however, there is still a lack of key tourist statistics in 2019 and some valuable data in 2020 after the epidemic. As the statistics from 2014 to 2018 are far away from the time node of the negative impact of the COVID-19 pandemic, and the impact of the new epidemic is changing rapidly, significant changes can occur within a few days. The most critical problem is that you lack key data within a period of time (maybe one month) before and after the outbreak of COVID-19 pandemic. This is what you need to make up for. It is recommended that you adopt Big data methods to capture relevant valuable data.

Thirdly, did you consider the reliability and validity of the statistical data in the principal component analysis? I did not see a clear explanation in the Results part of your manuscript.

Discussion:

There is no obvious problem, and it is recommended to streamline the discussion.

Conclusions:

In the conclusion part, it is not clear what is the specific strategies of sustainability strategy for Taiwan’s tourism industry?

6. PLOS authors have the option to publish the peer review history of their article (what does this mean?). If published, this will include your full peer review and any attached files.

Reviewer #1: No

Reviewer #2: No

---

## [Author Response · Author response to Decision Letter 0]

5 Jan 2021

Response to Reviewers

PONE-D-20-33157：

The manuscript (Developing a sustainability strategy for Taiwan’s tourism industry after the COVID-19 pandemic) has certain scientific research value under the current background that the COVID-19 still affects the global tourism economy. This study applied two-stage data envelopment analysis and principal component analysis to investigate past statistics and explore the shopping patterns of tourists who travel to Taiwan. The research methods are appropriate, and the research conclusions are novel to a certain extent.

The review result of the manuscript is Major Revision. My suggestion to the authors:

Introduction: 

Firstly, why do you use two-stage data envelopment analysis (DEA) and principal component analysis (PCA)? Are there any obvious benefit of DEA and PCA in this study, and what are the drawbacks of other major methods in this study? Or put why you use these methods in the "Methodology part" for explanation. 

Response:

Thank you for point out this drawback, I have explain the reason of using DEA and PCA in the "Methodology" part. Please refer to the revised version, from line 171-180.

Secondly, the application value of the paper is clearly explained, but does this study have theoretical and academic value? Whether there are academic theoretical shortcomings that need to be remedied in the context of the Taiwan region regarding the economic recovery of tourism after the epidemic? The academic research question of the paper is not very clear? Are you trying to find out where is the highest consumption area in Taiwan, or the greatest contribution to the tourism industry in Taiwan? From my judgment, you may want to look for reasonable sustainability strategy for the tourism economy after the COVID-19, and those small questions ( the highest consumption area or the greatest contribution to the tourism industry) are only for this main research question, therefore, in the Introduction part, you need to concentrate on deriving your research question reasonably and explaining its theoretical significance (if there is an ideal theoretical significance), so that it seems to be more logical in the Introduction part, otherwise it may make readers confused.

Response:

Thank you for your suggestion, I have rewrite the "Introduction" part. Please refer to the revised version, from line 67-71.

Related Works:

Firstly, delete “The United Nations World Tourism Organization” (in line 83), because you've explained in line 56 exactly what UNWTO stands for.

Response:

Thank you for your opinion, I have delete it. Please refer to the revised version, line 24 and 45.

Secondly, the part (the Current status of the international tourism market) is actually quite repetitive with the Introduction part, so it is suggested to simplify this part or enrich the main content of this part into the Introduction part.

Response:

Thank you for your opinion, I have combined this part into the “Introduction part”. Please refer to the revised version, line 24-59.

Thirdly, “New vaccines may be developed soon, so there is hope that the epidemic will end within a few months. However, international tourists may not come back as soon as expected. Moreover, consumer behavior may change temporarily or even forever. Therefore, how to attract international tourists to visit Taiwan again as soon as possible is a very important question, and the country must focus on those tourists who are most helpful to their operational performance so that the industry can recover in the shortest time possible.” (in line 137-141) 

Whether these above contents have basis or document source, or just your guess and judgment.

Response:

Thank you for your opinion, I have rewrite this part. Please refer to the revised version, line 140-144.

Methodology: 

It is recommended to make a table or figure to illustrate the benefits of the two methods (in fact, you had mentioned in your literature review, but they look scattered.), and point out why it is suitable to analyze statistics of visitors to Taiwan from the Tourism Bureau, so that the characteristics of the research methods ( DEA-CCR model and PCA) will be more clear.

Response:

Thank you for your suggestion, I have rewrite the "Methodology" part. Since there are already many tables in this article, I try to explain the reason of using DEA and PCA in the text. Hope you can agree this. Please refer to the revised version, line 159-180.

Results: 

Firstly, the part of “Two-stage DEA analysis” (in line 247-263) is recommended to be put into the "Methodology" part, because it is redundant to put it in the "Results" part.

Response:

Thank you for your opinion, I have move this part into the "Methodology" part and rewrite the methodology part. Please refer to the revised version, line 182-216.

Secondly, you should know that COVID-19 is probably confirmed by outstanding medical scientists around the world and may have existed before December 2019, but COVID-19 really obviously negative affects the global tourism industry after January 2020. and it is possible that the Taiwan region will be affected by the obvious negative impact of the new crown epidemic after January 2020. And you used the DEM method to analyze the data from 2014 to 2018. Although it can reflect the distribution and consumption of tourists to Taiwan during the period before the new crown epidemic to a certain extent, there is still a lack of key tourist statistics for 2019. You used the Two-stage DEA method to analyze the data from 2014 to 2018, although it can reflect to a certain extent the distribution and consumption of tourists who visited Taiwan during the period before the COVID-19 pandemic, however, there is still a lack of key tourist statistics in 2019 and some valuable data in 2020 after the epidemic. As the statistics from 2014 to 2018 are far away from the time node of the negative impact of the COVID-19 pandemic, and the impact of the new epidemic is changing rapidly, significant changes can occur within a few days. The most critical problem is that you lack key data within a period of time (maybe one month) before and after the outbreak of COVID-19 pandemic. This is what you need to make up for. It is recommended that you adopt Big data methods to capture relevant valuable data.

Response:

Thank you for your suggestion, I have corrected all data to 2019 or the latest available in 2020. Please refer to the revised version, table 1-8 and line 134-135.

Thirdly, did you consider the reliability and validity of the statistical data in the principal component analysis? I did not see a clear explanation in the Results part of your manuscript.

Response:

Thank you for your suggestion, I have cite more relative research as reference about PCA and rewrite the "Results" part. Please refer to the revised version, line 222-224, and the "Results" part, line 371-414.

Discussion: 

There is no obvious problem, and it is recommended to streamline the discussion. 

Response:

Thank you for your suggestion, I have rewrite the "Discussion" part. Please refer to the revised version, line 416-465.

Conclusions:

In the conclusion part, it is not clear what is the specific strategies of sustainability strategy for Taiwan’s tourism industry?

Response:

Thank you for your suggestion, I have mentioned these in the "Discussion" and "Conclusion" part. Please refer to the revised version.

English language: 

It is noted that your manuscript needs careful editing by someone with expertise in technical English editing paying particular attention to English grammar, spelling, and sentence structure so that the goals, process and results of the study are clear to the reader. Some sentences contain grammatical and/or spelling mistakes. But I believe the authors can make reasonable corrections.

Response:

Thank you for your suggestion, this article have been English edited by expertise. However, I will do it again if necessary.

---

## [Decision Letter · Decision Letter 1]

24 Feb 2021

Developing a sustainability strategy for Taiwan’s tourism industry after the COVID-19 pandemic

PONE-D-20-33157R1

Dear Dr. Tsai,

We’re pleased to inform you that your manuscript has been judged scientifically suitable for publication and will be formally accepted for publication once it meets all outstanding technical requirements.

Kind regards,

Bing Xue, Ph.D.

Academic Editor

PLOS ONE

Additional Editor Comments (optional):

Reviewers' comments:

Reviewer's Responses to Questions

**Comments to the Author**

1. If the authors have adequately addressed your comments raised in a previous round of review and you feel that this manuscript is now acceptable for publication, you may indicate that here to bypass the “Comments to the Author” section, enter your conflict of interest statement in the “Confidential to Editor” section, and submit your "Accept" recommendation.

Reviewer #2: All comments have been addressed

2. Is the manuscript technically sound, and do the data support the conclusions?

Reviewer #2: Partly

3. Has the statistical analysis been performed appropriately and rigorously? 

Reviewer #2: I Don't Know

4. Have the authors made all data underlying the findings in their manuscript fully available?

Reviewer #2: Yes

5. Is the manuscript presented in an intelligible fashion and written in standard English?

Reviewer #2: Yes

6. Review Comments to the Author

Reviewer #2: Congratulations, and strive to make this research problem clearer next time and have more supporting data.

7. PLOS authors have the option to publish the peer review history of their article (what does this mean?). If published, this will include your full peer review and any attached files.

Reviewer #2: No

---

## [Editor Report · Acceptance letter]

1 Mar 2021

PONE-D-20-33157R1 

Developing a sustainability strategy for Taiwan’s tourism industry after the COVID-19 pandemic 

Dear Dr. Tsai:

I'm pleased to inform you that your manuscript has been deemed suitable for publication in PLOS ONE. Congratulations! Your manuscript is now with our production department. 

Kind regards, 

on behalf of

Professor Bing Xue 

Academic Editor

PLOS ONE